# Interstitial Lung Disease and Pulmonary Fibrosis: A Practical Approach for General Medicine Physicians with Focus on the Medical History

**DOI:** 10.3390/jcm7120476

**Published:** 2018-11-24

**Authors:** Or Kalchiem-Dekel, Jeffrey R. Galvin, Allen P. Burke, Sergei P. Atamas, Nevins W. Todd

**Affiliations:** 1Department of Medicine, University of Maryland School of Medicine, Baltimore, MD 21201, USA; satamas@som.umaryland.edu (S.P.A.); ntodd@som.umaryland.edu (N.W.T.); 2Department of Radiology and Nuclear Medicine, University of Maryland School of Medicine, Baltimore, MD 21201, USA; jgalvin@umm.edu; 3Department of Pathology, University of Maryland School of Medicine, Baltimore, MD 21201, USA; aburke@umm.edu; 4Baltimore Veterans Affairs Medical Center, Baltimore, MD 21201, USA

**Keywords:** smoking, hypersensitivity, autoimmune, occupation, idiopathic pulmonary fibrosis

## Abstract

Interstitial lung disease (ILD) and pulmonary fibrosis comprise a wide array of inflammatory and fibrotic lung diseases which are often confusing to general medicine and pulmonary physicians alike. In addition to the myriad of clinical and radiologic nomenclature used in ILD, histopathologic descriptors may be particularly confusing, and are often extrapolated to radiologic imaging patterns which may further add to the confusion. We propose that rather than focusing on precise histologic findings, focus should be on identifying an accurate etiology of ILD through a comprehensive and detailed medical history. Histopathologic patterns from lung biopsy should not be dismissed, but are often nonspecific, and overall treatment strategy and prognosis are likely to be determined more by the specific etiology of ILD rather than any particular histologic pattern. In this review, we outline a practical approach to common ILDs, highlight important aspects in obtaining an exposure history, clarify terminology and nomenclature, and discuss six common subgroups of ILD likely to be encountered by general medicine physicians in the inpatient or outpatient setting: Smoking-related, hypersensitivity pneumonitis, connective tissue disease-related, occupation-related, medication-induced, and idiopathic pulmonary fibrosis. Accurate diagnosis of these forms of ILD does require supplementing the medical history with results of the physical examination, autoimmune serologic testing, and chest radiographic imaging, but the importance of a comprehensive environmental, avocational, occupational, and medication-use history cannot be overstated and is likely the single most important factor responsible for achieving the best possible outcomes for patients.

## 1. Introduction

Interstitial lung disease (ILD) and pulmonary fibrosis comprise a wide array of lung diseases which are often confusing to general medicine and pulmonary physicians alike. One of the confusing areas in ILD is the myriad of terms, abbreviations, and acronyms that are applied to imaging patterns, histopathology findings, and integrated clinical diagnoses. Our goals in this review are to highlight the importance of a comprehensive medical history, to provide a practical framework that promotes understanding of major etiologic groups of ILDs, to enhance clarity regarding current nomenclature, and to give a brief overview of available treatment options for specific ILDs. We will attempt to use ILD terms concisely and define them clearly. We will focus on six of the most common and important subacute and chronic forms of ILD likely to be seen by general medicine physicians in the inpatient or the outpatient setting: Smoking-related, connective tissue or autoimmune disease-related, hypersensitivity pneumonitis, occupation-related, medication-induced, and idiopathic pulmonary fibrosis (IPF). Most ILD patients will likely ultimately be referred to a pulmonary physician for further evaluation, but our goal is to provide a framework for identifying the etiology of ILD and to provide a better understanding of the myriad of ILD terms and nomenclature to assist with the overall care of patients by general medicine physicians.

## 2. ILD Overview 

ILD and pulmonary fibrosis are a group of lung diseases which consist of a combination of inflammation and fibrosis of the lung parenchyma (Figure 1). There are many diverse causes of ILD, which usually result from a variety of environmental, avocational, occupational, or medication-related exposures, or alternatively may result from one of the numerous systemic autoimmune or connective tissue diseases (CTD) [1,2,3]. One particular form of ILD is termed IPF, and IPF is often considered one of the most common and important ILDs due to its unknown etiology, its poor overall prognosis, and its modest response to therapeutic interventions [4,5,6]. Therapy for ILD and pulmonary fibrosis may be complex at times, but will almost always be based in principle on the most likely etiology of ILD. Given the role of exposures in ILD and the significance of identifying a precise etiology, the importance of a detailed and comprehensive environmental, avocational, occupational, and medication-use history cannot be overstated, and is likely the most important factor in determining an accurate ILD diagnosis [7,8]. An accurate ILD diagnosis additionally requires supplementing the medical history with results of the physical examination, autoimmune serologic testing, chest computed tomography (CT) imaging, and lung biopsy if appropriate. Pulmonary function testing (PFTs) are essential to management and usually show a pattern of a restrictive ventilatory defect with an abnormal diffusion capacity, but are predominantly a guide to severity of disease and response to treatment, and only rarely will point to a specific form of ILD [1,9]. In selected instances, one piece of clinical information may supersede all others in arriving at accurate diagnosis, but most often, integration of the medical history in conjunction with serology, imaging and histology findings is essential in order to establish a precise diagnosis and guide therapeutic decisions [2,6,10,11].

Terminology of ILD along with its abbreviations and acronyms is often confusing to clinicians and some of this ambiguity and impreciseness is likely a reflection of an incomplete understanding of many of these disease processes. The term interstitial lung disease itself is also somewhat imprecise. Conditions classified as ILD should be pathobiologically restricted to the interstitium of the lung, which consists of alveolar septa and structures residing within these septae that exist between alveolar air spaces throughout the lung parenchyma. However, many diseases classified as ILD often involve the alveolar air space as well as the interstitium [8,12], making this distinction difficult and somewhat arbitrary. Alternative terms for ILD have included diffuse parenchymal lung disease and diffuse lung disease, however, these terms are probably just as imprecise and have not been as widely adopted. 

## 3. Pathophysiology

ILD is recognized as a group of diseases characterized by a combination of (a) chronic inflammation within the lung, consisting of an accumulation of chronic inflammatory cells (predominantly lymphocytes and macrophages) and increased levels of numerous pro-inflammatory cytokines, chemokines, and cell surface molecules; and (b) varying degrees of lung fibrosis [13,14]. Therefore, as shown in Figure 1, any particular form of ILD may manifest primarily as an inflammatory lung disease, with little if any features of fibrosis and a generally favorable response to anti-inflammatory or immunosuppressive treatments; may manifest primarily as a fibrotic pulmonary disease, with a generally poor prognosis and limited efficacious therapies; or may manifest with varying degrees of inflammation and fibrosis. The one exception is IPF, since IPF is considered a prototypical fibrotic lung disease (Figure 1), with anti-inflammatory or immunosuppressive treatments being ineffective and at times harmful. The term fibrosis generally implies histologically an excess deposition of collagen [13,15], but there have been conflicting reports over many years as to the precise histopathologic components of clinically recognized pulmonary fibrosis [16,17,18]. Despite this uncertainty, pulmonary fibrosis can reliably be recognized on CT imaging of the chest by its characteristic findings (see Radiology section below). 

## 4. Comprehensive Medical History

As mentioned above, the importance of a detailed and comprehensive environmental, avocational, occupational, and medication-use history is likely the most important factor in determining an accurate ILD diagnosis. Detailed questioning needs to be directed towards the following: Inhaled substance use, including cigarettes, cigars, marijuana, cocaine, or other inhaled illicit drugs; the home environment, with particular attention to water intrusion, presence of visible mold, prior remediation efforts for water damage or mold infestation, or water reservoirs inside or in vicinity of the home environment which may contain mold; regular exposure to birds or avian proteins as either pets or as part of an occupation or hobby in a variety of environments; other hobbies, including those involving wind musical instruments, ceramics, other dusty environments, or animals; a comprehensive occupational history, including exposures to any of the well-known pulmonary toxins (e.g., asbestos), dusts, gases, or fumes, and whether respiratory protective gear was used or required; and a comprehensive medication-use history, including herbal medications and any therapies used for malignancy treatment [7,8]. Additionally, a detailed family history is very informative in patients with ILD. Over the past several decades, a genetic component to ILD and pulmonary fibrosis has been recognized in selected patients and families. Nomenclature in this area is somewhat imprecise, and includes terms such as familial ILD, familial pulmonary fibrosis, familial interstitial pneumonia, and familial IPF. Two broad groups of genes identified in families with inherited disease are those related to telomere biology [19] and surfactant protein processing [20,21].

It should be noted that even for the most experienced clinicians, obtaining this history is often a time-consuming endeavor, and thus ILD questionnaires that can be completed by patients to supplement physician questioning are often very useful, and many are available in the public domain from a number of medical institutions and international organizations, such as the American College of Chest Physicians (ACCP) [22].

One other consideration based on the comprehensive medical history is the possibility of an infectious pulmonary process. In general, the numerous terms and abbreviations used in ILD nomenclature refer to non-infectious processes in the lung parenchyma. In selected instances, particularly in the presence of immunosuppression, pulmonary infections caused by mycobacterial (e.g., *Mycobacterium avium*) [23], viral (e.g., cytomegalovirus) [24], or fungal (e.g., *Pneumocystis jiroveci*) pathogens [25] may mimic non-infectious ILD. Careful consideration of the medical history and physical examination for symptoms and signs of infection is thus always warranted, and bronchoscopy with bronchoalveolar lavage (BAL) and/or biopsy may be considered as further evaluation for potential infectious pathogens in selected patients.

## 5. Histology

To general medicine physicians and to many pulmonary physicians as well, the terms and abbreviations relating to the histopathology of ILD are confusing [26]. Our goal is to introduce and explain a few of the most common terms so these can be recognized by internists in histopathology reports, and to provide some framework in this area. 

Three major histopathologic patterns which occur in response to lung injury in ILD are usual interstitial pneumonia (UIP), non-specific interstitial pneumonia (NSIP), and organizing pneumonia (OP), and these are illustrated in Figure 2, panels (A–F). One can clearly note the overlapping terminology and multiple acronyms. UIP is recognized microscopically by heterogeneous areas of dense fibrosis interspersed with areas of relatively normal lung architecture [4,6], the presence of honeycomb change, which consists of the presence of numerous microscopic cysts in the lung parenchyma which are lined by mature respiratory epithelium and filled with mucus, and the presence of fibroblastic foci, which are microscopic oval or round areas of extracellular matrix and fibroblasts often juxtaposed to honeycomb cysts. NSIP is recognized microscopically by diffuse cellular inflammation and/or fibrosis in the lung interstitium that occurs in a spatially homogenous pattern throughout the lung, and in which overall lung architecture is preserved [27,28]. OP, or area of organization, is recognized microscopically by multiple round or oval pale-staining deposits consisting of extracellular matrix proteins and spindle-shaped fibroblasts or myofibroblasts [12,29,30]. Note the distinguishing features of UIP and NSIP, in which UIP shows heterogeneous areas of dense fibrosis interspersed with normal lung, whereas NSIP shows spatially uniform injury throughout the lung [5,31]. Although there may be overlap, these three patterns can generally be distinguished by a pathologist with clinical experience in the histopathology of ILD. Granulomatous inflammation is an additional pattern of lung injury, which consists of characteristic aggregates of lymphocytes and histiocytes, and often accompanied by the presence of multinucleated giant cells, and represented in Figure 2, panels (G,H). Granulomatous inflammation is characteristic of hypersensitivity pneumonitis (HP), as discussed later in this review, and additionally is observed in sarcoidosis. Granulomas in sarcoidosis generally tend to be well-formed and often demarcated by surrounding concentric rims of fibrosis, whereas granulomas in HP are most often poorly-formed. A few further specific histopathology findings which are found in smoking-related ILD will be discussed in its specific section. 

Although identifying a histopathologic pattern in ILD may be helpful, an individual patient’s treatment strategy and prognosis are likely to be more determined by the specific etiology of ILD rather than a specific histologic pattern. Each of the histopathologic patterns UIP, NSIP, and OP can be seen in numerous diverse ILDs, many of which have markedly different prognoses and markedly different responses to therapy [1,2,3]. As an example, UIP is the characteristic histopathologic pattern observed in IPF, a disease with uniformly poor prognosis and uniformly poor response to anti-inflammatory therapy [4]. However, a UIP histopathologic pattern can also be observed in patients with connective tissue disease-related ILD, and these patients generally will have a better prognosis and better response to anti-inflammatory therapy compared to IPF patients despite an identical histopathologic pattern [32]. These observations suggest that patterns of lung injury are nonspecific and reinforce the concept that treatment response and clinical prognosis are likely determined more by the etiology and temporal nature of lung injury (ceased versus persistent) rather than a specific histopathologic pattern. 

Though a bronchoscopic lung biopsy is very useful in malignant and infectious pulmonary diseases, it is generally not helpful in most forms of ILD due to the small size of the biopsies [6], with a few diseases such as sarcoidosis being an exception [33]. Therefore, whether a patient with ILD should have a surgical lung biopsy (SLB) is often one of the most debated and controversial topics in pulmonary medicine. Advantages of SLB are precisely defining the histopathologic pattern, but as mentioned above, treatment and prognosis in ILD are more likely to be determined by the etiology of ILD rather than a specific histologic pattern. Disadvantages of SLB are related to risks of the procedure, and in addition to the usual risks of an invasive operative procedure, a substantial deterioration in pulmonary status or even death may occur following SLB in various forms of ILD secondary to what is termed “acute exacerbation of ILD” [34,35]. The decision to proceed with SLB is thus always individualized, taking into account risks versus benefits, co-morbidities, and patient preferences [2,6,36].

## 6. Radiology

Radiologic patterns on chest CT imaging are an essential supplement to a comprehensive medical history for making an accurate ILD diagnosis. As with histopathology of ILD, chest CT imaging findings have their own vocabulary, and likely adding to ILD complexity is that histopathology terminology is at times extrapolated to chest CT imaging findings. For example, chest CT imaging patterns may be described as representing a UIP, NSIP, or OP pattern based on the likely histopathologic appearance of the lung [37]. 

Three features of chest CT imaging deserve specific mention and are shown in Figure 3 and Figure 4. First, the distribution of opacities within the lung parenchyma needs to be identified: (a) upper lobe-versus lower lobe-predominance; (b) peripheral versus central (central is also termed peribronchiolar); and (c) whether the opacities spare the extreme periphery of the lung. A precise explanation for these varying distributions is often not known, but specific forms of ILD have characteristic distributions of opacities on CT imaging. As two examples, hypersensitivity pneumonitis is typically an upper lobe-predominant disease [38], whereas IPF is typically a lower lobe-predominant disease [4,6]. Coronal and sagittal reformatted images, which supplement standard axial images, greatly assist with recognizing the distribution of disease. Second, the chest CT findings of fibrosis need to be recognized, and there are four findings which usually occur in combination which indicate the presence of pulmonary fibrosis: (a) reticular opacities, which are small curvilinear lines often located in the periphery of the lung; (b) traction bronchiectasis, which represents dilated, distorted, and irregularly-shaped bronchi and bronchioles that develop due to contracting areas of fibrosis in the surrounding alveolar structures; (c) honeycomb change, analogous to the histopathology descriptor, which on radiology is observed as clusters of small cysts located in the extreme periphery of the lung, likely representing dilated distal ends of small bronchioles [37,39] and so-named due to the resemblance to a bee honeycomb; and (d) volume loss, which occurs as fibrosis results in loss of tissue aeration and volume, and can be most easily be recognized by abnormal displacement of the major and minor fissures within the lung. It should be noted that traction bronchiectasis [40,41] and honeycombing [42] likely represent advanced stages of fibrosis and portend a poor prognosis regardless of the underlying etiology. Third, the presence of ground glass opacities (GGO) and/or consolidation should be recognized. GGO is an area of mildly hyperattenuated lung in which the underlying vasculature is still visible, whereas consolidation is an area of densely hyperattenuated lung in which patent small airways coursing through the consolidated area, termed air bronchograms, are often visible. GGO and consolidation tend to be considered as inflammatory in nature, and potentially more reversible than the findings seen with fibrosis [37]. 

## 7. ILD Subgroups

In the following section we will review six common subgroups of ILD. Distinguishing clinical, radiographic, laboratory, and histopathologic characteristics among these six subgroups are summarized in Table 1.

### 7.1. Smoking Related-Interstitial Lung Disease 

Over the past 10 to 15 years, it has become well-established that inhalation of tobacco smoke or other smoke inducing products is associated with a variety of forms of ILD [43]. Although tobacco smoking is the most common cause of emphysema and chronic obstructive pulmonary disease (COPD), various other types of lung injury have been described due to smoke exposure and are encompassed within the ILD spectrum of disease [44,45,46]. These include, along with their own specific terminology and acronyms, respiratory bronchiolitis (RB), desquamative interstitial pneumonia (DIP), pulmonary Langerhans cell histiocytosis (PLCH), smoking-related interstitial fibrosis (SRIF), and combined pulmonary fibrosis and emphysema (CPFE). The precise amount of smoke exposure required to cause the various forms of ILD is unknown, but is generally thought to occur in a dose-dependent fashion [47,48]. A detailed history regarding the use and quantity of cigarettes, electronic cigarettes, marijuana, or other illicit substances in conjunction with chest CT imaging patterns will indicate smoking-related ILD as the most likely specific process.

RB and DIP are usually considered spectrums of the same disease process [1,2,49] and are defined based on radiographic and histopathologic findings. Both disorders demonstrate the abnormal accumulation of finely pigmented alveolar macrophages in the lung as a result of smoke inhalation. In RB, pigmented macrophages accumulate focally surrounding small airways [50], whereas in DIP, pigmented macrophages accumulate more diffusely throughout the lung parenchyma [51]. Radiologically, both RB and DIP usually manifest as GGO, though in differing distributions and severity. RB usually consists of ill-defined ground-glass nodules of varying size (1–10 mm) generally in an upper lung zone distribution, whereas DIP is generally characterized by more extensive GGOs, tends to be more pronounced in the lower lobes, and may have a fibrotic component manifested by traction bronchiectasis and volume loss (Figure 3A) [45,51]. It should be noted that the term “desquamative”, coined in the 1960s and still in use today, was chosen since these abnormal cells were originally thought to represent desquamated epithelial cells [52]. However, based on our current understanding of RB and DIP, this term is thus a misnomer since these abnormal cells are in reality pigmented alveolar macrophages. 

PLCH is recognized by distinctive radiographic and histopathologic findings in a patient with a history of smoking. Histopathologic findings of PLCH include stellate shaped nodules with chronic inflammatory cell accumulation, the hallmark of which is the accumulation of CD1a-positive histiocytes referred to as Langerhans cells [53]. Radiographically, the findings of PLCH occur in an upper lobe-predominant distribution and are often striking, consisting of a combination of various sized lung nodules and bizarre-shaped cysts, both of which give PLCH its distinct appearance [45].

SRIF refers to emphysematous air spaces with abnormally thick fibrotic walls. Typically, straight-forward upper lobe-predominant centrilobular emphysema secondary to smoking presents radiographically as enlarged air spaces with no true discernable walls. In SRIF, the walls of these emphysematous spaces are abnormally thick, well-defined, and contain excess collagen, thus being most consistent with smoking-induced fibrosis which surrounds emphysematous spaces [54,55]. This process has also been referred to as airspace enlargement with fibrosis, which seems descriptively more accurate [2], but smoking-related interstitial fibrosis (SRIF) seems to have gained more acceptance [56]. 

Over the last decade, one other entity related to smoking-induced lung injury which has gained attention has been termed combined pulmonary fibrosis and emphysema (CPFE) [57]. Though somewhat non-specific in terms of distribution of disease, this term has generally been used to describe centrilobular emphysema in the upper lobes and lung fibrosis in the lower lobes [58]. Of note, in SRIF described above, the fibrosis component is observed in a similar distribution as the emphysema [54,55], and it thus seems straight forward to conclude that tobacco smoke is a direct cause of the fibrotic change [59]. With upper lobe emphysema and lower lobe fibrosis, such as in CPFE, it is difficult to attribute the fibrotic change with certainty to tobacco smoke. Thus, many of these patients termed CPFE may in fact represent merely a variant of IPF, since most patients diagnosed with IPF have a prior history of tobacco smoking [60]. 

The first-line therapeutic approach for all forms of lung injury due to tobacco smoking or other inhalants is smoking cessation [56]. In patients with progressive disease, consideration of anti-inflammatory therapy with systemic corticosteroids is appropriate, though efficacy of such treatment for any of the smoking-related ILD forms has not been convincingly demonstrated [49,61].

### 7.2. Hypersensitivity Pneumonitis

HP, also referred to as extrinsic allergic alveolitis, is one of the most important ILDs to recognize due to possible environmental interventions or remediations which could provide substantial patient benefit [62]. HP results from an exuberant immunologic response to inhalation of environmental organic antigens [63], originating most commonly from exposure to avian proteins [64], mold [65], or farming [66,67]. Some forms of HP may result from inhalation of inorganic antigens [68,69], but most implicated antigens are organic in nature. Numerous and at times colorful descriptor terms have been used to describe HP due to a variety of organic exposures, such as hot tub lung, pigeon breeders’ lung, or cheese workers’ lung.

The recognition and identification of the etiology of HP rests almost solely on a detailed environmental, avocational, and occupational history [70]. Specific attention needs to be directed at the possibility of mold infestation in the home or work environment [71]. Clues in this regard include a history of water intrusion into the premises, damp or musty spaces, musty odor, or visible mold infestation. In some instances, inspection of the premises by a certified industrial hygienist will allow accurate assessment as to the presence of mold infestation and provide guidance for potential remediation if necessary [72]. Additionally, patient-directed questions regarding exposure to avian proteins are essential, and should assess not only birds residing within the household, but potentially frequent contact with birds or bird excrement outside of the home such as in coops, chicken houses, and other similar dwellings [73,74]. Different panels of serum precipitating antibodies directed at common environmental and avian antigens are commercially available, and although their utility remains a source of debate due to low sensitivity and specificity, a positive test in the appropriate clinical setting helps support the diagnosis of HP [63]. It is important to note that in some instances, despite a comprehensive environmental, avocational, and occupational history by experienced physicians, identification of a precise antigen in patients with clinical presentation highly suggestive of HP may be difficult [75,76].

HP can present in an acute, subacute, or chronic manner, but often manifests as insidious ILD of uncertain etiology. In this instance, radiographic findings are often present in an upper lobe-predominant fashion, and often accompanied by typical findings of fibrosis [38,77]. A mosaic attenuation CT pattern is also supportive of chronic HP (Figure 3B), in which areas hyperattenuated lung are interspersed with areas of hypoattenuated lung, and likely represents sequela of small airways injury and focal air trapping. The hypoattenuated areas are often accentuated when expiratory imaging protocols are additionally applied [77]. Histopathologic findings in HP consist of chronic inflammation centered around small airways, and the presence of poorly-formed granulomas (Figure 2, panels (G,H)). Poorly-formed granulomas noted on a lung biopsy specimen should always prompt consideration of HP as a diagnosis. 

Therapeutic interventions for chronic HP involve most importantly cessation of the inciting exposure [78]. In the case of avian-related HP, complete cessation of all exposure to avian proteins is crucial [79], and a thorough professional cleaning of the environment needs to be undertaken if the patient is to remain within that environment [80]. It should be noted that even following professional cleaning, significant avian products may remain detectable in the environment [64]. In the case of mold, remediation of the environment in attempt to cease water intrusion and remove mold infested materials is of utmost importance [72]. This work should be performed by a professional company with remediation expertise since significant antigen dispersion into the environment may occur during this process. Regarding other organic antigens from different environmental exposures, the same principles of exposure cessation and antigen removal from the patient’s environment are essential [78].

The role of corticosteroids and other immunosuppressive agents in chronic HP is not completely defined. Numerous reports do suggest improvement in clinical status and PFTs with corticosteroids [81,82], though other reports have indicated minimal to no improvement [83]. Likewise, the role of alternative immunosuppressive therapies such as mycophenolate mofetil or azathioprine remains to be proven at the present time [81,84]. Some reports suggest that response to anti-inflammatory therapy in particular patients may be predicted based on radiographic features which are present or absent on chest CT imaging at presentation [85].

### 7.3. Connective Tissue Disease-Associated ILD

Numerous systemic connective tissue diseases (CTD) and autoimmune conditions have been associated with ILD. Although there may be strict differences between CTD and autoimmune disease, we will use these terms interchangeably in this review. It is particularly important to identify CTD disease as the inciting ILD etiology due to the high likelihood of physiologic improvement with corticosteroids and other immunosuppressive agents [32,86]. CTDs commonly associated with ILD include systemic lupus erythematosus (SLE), systemic sclerosis (SSc), Sjögren’s syndrome, rheumatoid arthritis (RA), mixed connective tissue disease (MCTD), systemic vasculitis, and the anti-synthetase syndromes, including the inflammatory myopathies, polymyositis (PM) and dermatomyositis (DM) [87]. Additionally, some patients with ILD have concomitant autoimmune phenomena that cannot be classified clinically and/or serologically under one of the well-established CTD or autoimmune diseases [88,89,90]. The American Thoracic Society and European Respiratory Society have addressed this entity by designating it interstitial pneumonia with autoimmune features [91]. The pathophysiology of CTD–ILD likely involves repetitive and progressive lung injury due to systemic inflammation [92], and the pathophysiologic role of a secondary insult from inhalation of various substances remains to be discerned [93,94].

The recognition and identification of patients with CTD-ILD occurs mostly from a constellation of the medical history, physical exam, radiologic, and serologic findings. A history of constitutional symptoms, arthralgias, arthritis, rashes, and other systemic symptoms may suggest the possibility of an underlying CTD, and physical exam findings specific to a particular CTD are often very helpful [86,95,96,97]. In the appropriate clinical context, serologic markers of autoimmunity are very helpful in understanding the likely pathobiology of the ILD. It has been somewhat controversial in the past as to whether all patients with ILD should undergo a full CTD serologic evaluation looking for evidence of autoimmune disease [91,98]; most consensus statements do support this approach, and it is our belief as well that almost all patients with ILD should undergo such testing as it can have profound impact on therapeutic management and prognosis [2,4,6,91]. Table 2 shows a list of suggested autoimmune serology and additional assays that can be considered in the work-up of patients with ILD. The choice of assays may be individualized based on the patient’s presentation and the clinical context [6].

Radiologic findings in CTD-ILD are nonspecific [42,89,97], and usually consist of a combination of reticular opacities, GGO, consolidation, and varying degrees of fibrosis (Figure 3, panels (C,D)). The opacities often have a pattern in which the extreme periphery of the lung is spared (Figure 3D) [99]. Most patients with CTD-ILD will not require a lung biopsy, since treatment response and prognosis are likely to be determined more by the presence of ILD secondary to CTD rather than the specific histopathologic pattern that happens to be present [32,100,101,102,103]. In selected instances, bronchoscopy with BAL may be useful to exclude infection and to evaluate the degree of active alveolitis [104,105]. If a lung biopsy is performed, any of the typical histopathologic patterns seen in ILD including UIP, NSIP, or OP may be observed in varying amounts and distributions [106,107].

Of the utmost importance in CTD-ILD is therapy with corticosteroids and/or other immunosuppressive agents. Arguably, patients with ILD secondary to CTD have the best responses to immunosuppressive therapy compared to other ILD groups [32,87,100,101,102,103,108,109]. Corticosteroids have long been the mainstay of immunosuppressive therapy for patients with CTD-ILD [110,111], however clinical experience and research studies have shown additional benefits of steroid-sparing agents, such as azathioprine [112,113,114], methotrexate [115,116], cyclophosphamide [117,118,119,120], and more recently, mycophenolate mofetil [113,119,120,121,122]. Over the past five years, mycophenolate mofetil has gained widespread acceptance for use in patients with CTD-ILD as it is generally well-tolerated and efficacious [113,118,119,120,121,122]. 

One syndrome in patients with CTD-ILD that warrants particular attention is the anti-synthetase syndrome. This syndrome presents as one of the inflammatory myopathies—PM or DM—in combination with the presence of one or more anti-aminoacyl-tRNA synthetase antibodies on serologic testing, and the presence of radiographic ILD [123]. It is important to note that many of these patients may not have overwhelming clinical signs of systemic disease [124], and it has been well described that many patients may not have overt evidence of myositis either clinically or by serum muscle enzyme testing [124,125,126,127]. Serology panels of myositis or anti-synthetase antibodies are now commercially available, and usually include 10 or more antibodies that are either specific (myositis-specific) or associated (myositis-associated) with the anti-synthetase syndrome (Table 2) [128,129]. In general, patients with ILD and anti-synthetase antibodies respond well to corticosteroids and other immunosuppressive therapies, however most experts would state that corticosteroid therapy alone is insufficient to control and improve the disease process in most patients [96,102]. One particular anti-synthetase antibody associated with severe ILD is melanoma differentiation-associated gene (MDA)-5, also known as clinically amyopathic dermatomyositis (CADM)-140. In these MDA-5-positive patients, aggressive immunosuppression therapy is usually warranted, and either intravenous immunoglobulin and/or therapy with rituximab have been shown to be helpful in these patients [126,130,131,132].

### 7.4. Occupation-Related ILD

Traditionally, occupation-related lung diseases were some of the earliest recognized forms of ILD. Occupations involving exposure to coal, silica, and asbestos have long been recognized as causing lung disease classified within the ILD spectrum, and the term pneumoconiosis has been applied here as well (e.g., coal miners’ pneumoconiosis) [133,134,135]. Over the past several decades, many other occupations and substances have been grouped under occupation-related lung disease, and include exposures to beryllium, cobalt, tungsten, aluminum, and other chemicals [136,137].

Similar to HP, the recognition and identification of occupation-related ILD rests almost solely on a detailed occupational history. Patient questionnaires which require recording of lifetime occupations in detail may assist with identifying potential exposures [138,139], and there are a variety of questionnaires available in the public domain from medical institutions and international societies, such as the previously-mentioned ACCP questionnaire [22], among others [140]. 

From a radiologic standpoint, imaging findings in occupation-related ILD may suggest a particular exposure, particularly when coupled with the patient’s occupational history, or may be nonspecific. Exposure to coal and silica tend to result in nodular upper lobe-predominant opacities with varying degrees of calcified or non-calcified mediastinal adenopathy [141,142]. ILD as a result of asbestos exposure generally results in lower lobe-predominant opacities similar or even indistinguishable from IPF, but the presence of bilateral calcified pleural plaques is a relatively specific indicator of prior exposure, and their presence would strongly suggest an asbestos etiology for the ILD [143]. Beryllium is a metal compound widely found in the aerospace and defense industries as well as nuclear facilities and alloy manufacturing, among others. Chronic beryllium disease radiographically closely mimics sarcoidosis (Figure 3E) [144,145] and a diagnosis of sarcoidosis in any patient should always prompt the clinician to at least enquire about possible beryllium exposure. Radiographic patterns in ILD secondary to other occupational exposures are varied, but usually nonspecific to any particular exposure, underscoring the importance of the medical history in occupation-related ILD.

Prevention is the cornerstone of management of occupation-related morbidity by eliminating or minimizing exposure to hazardous emissions, imposition of stringent engineering controls, use of respiratory protective gear, and vigorous employee compliance with the above-mentioned strategies [146]. As with HP, removal of the patient from the exposure and cessation of further exposures is crucial in management in occupation-related ILD [147]. Unfortunately, in some instances, severe fibrotic lung disease is not recognized until after many years of continuous exposure, or not recognized until the patient is no longer in the workforce and no longer exposed to the offending agent. Therapies for established occupation-related ILD are very limited. Corticosteroids in general are not efficacious in coal-, silica-, or asbestos-induced ILD [148], but may have some efficacy in symptomatic chronic beryllium disease [149,150].

### 7.5. Medication-Induced ILD

ILD and pulmonary fibrosis secondary to medication toxicity has likewise long been recognized, and numerous medications over many years have been identified as causes of lung injury and pulmonary toxicity. Some of the earliest medications associated with pulmonary toxicity have faded over time from current use for a variety of reasons [151], but new pharmaceutical agents continue to be developed and their potential to induce pulmonary injury is sometimes recognized only in the post-marketing period with wider use of the medication [152]. Recognizing that ILD in a particular patient may be caused by a medication can be ascertained only by obtaining a careful medication history from the patient and from the medical record, since in most instances, it is unlikely that radiologic or histopathologic patterns will be specific for medication-induced pulmonary toxicity [153].

Several classification schemes exist to attempt to organize the numerous medications which have been associated with pulmonary toxicity. Some tend to center on the indications for clinical use and such examples include cardiovascular medications (e.g., amiodarone), antimicrobials (e.g., nitrofurantoin), biologics (e.g., tumor necrosis factor-inhibitors), neurologic medications (e.g., bromocriptine), herbal medications, and numerous anti-neoplastic medications. Classifications by acuity, radiographic pattern, or histopathologic pattern of lung injury are also commonly used. The open-source website Pneumotox On Line is a readily-available, user-friendly, and valuable resource in this regard [154], and this database is also available in application form for iPhone^TM^.

Pulmonary toxicity from external beam radiation therapy is often additionally considered within the context of adverse pulmonary reactions secondary to anti-neoplastic therapy. Pulmonary manifestations of radiation toxicity include radiation pneumonitis and radiation fibrosis [155]. Furthermore, radiation pneumonitis may be potentiated by concurrent or prior sensitizing anti-neoplastic medications [156].

Pulmonary toxicity due to medication use can present with varying patterns on chest CT imaging. Any combination of distributions and patterns, including GGO, nodular opacities, consolidation, and findings of fibrosis can be observed in selected instances (Figure 3F) [157]. In addition, pleural effusion and/or pleural thickening may be present in some cases of medication-related toxicity [158]. Likewise, the histopathologic pattern of medication-related toxicity is most often nonspecific, and any of the recognized patterns of lung injury including chronic inflammation, pulmonary hemorrhage, granulomatous inflammation, UIP, NSIP, and OP have all been observed with various medications over the years [159].

It seems appropriate to additionally mention two classes of drugs which are relatively new to the medical community whose indications are expanding, but have been observed to cause pulmonary toxicity and ILD in some patients. First are the proteasome inhibitors, which are used predominately in the treatment of hematologic malignancies, including multiple myeloma. These drugs likely exert their action by promoting a pro-apoptotic environment in malignant cells via inhibition of protein-degradation by the proteasome [160]. The three proteasome inhibitors currently approved for use by the US Food and Drug Administration (FDA) are bortezomib, carfilzomib, and ixazomib. As the initial agent approved, pulmonary toxicity associated bortezomib has been recognized with multiple different radiologic and histopathologic patterns [161]. Corticosteroids may be beneficial in patients who develop pulmonary toxicity secondary to proteasome inhibitors, though not uniformly so [162].

The second class of medications are the programmed cell death (PD)-1 and the PD-ligand (PD-L) 1 inhibitors. These medications group under the larger class of immune checkpoint inhibitors [163] and have been increasingly used over past several years for the treatment of non-small cell lung cancer and melanoma. PD-1 and PD-L1 are co-stimulatory molecules involved in antigen presentation to T-cells, and in the normal host, binding of antigen-presenting cell PD-L1 to T cell PD-1 receptor dampens the immune response. In selected instances, malignant cells have been demonstrated to express PD-L1 on their cell surface, and thus inhibition of either PD-1 or PD-L1 has been associated with immune system activation and improved outcomes [163]. Two PD-1 inhibitors, pembrolizumab and nivolumab, and three PD-L1 inhibitors, atezolizumab, avelumab, and durvalumab, have been approved by the FDA. As with the proteasome inhibitors, pulmonary toxicity associated with these medications has been recognized as their use has increased [164,165,166]. Many patients will improve with corticosteroid therapy, though relapse with corticosteroid tapering is common [167,168].

### 7.6. Idiopathic Pulmonary Fibrosis

IPF is often considered one of the most important ILDs due to its unknown etiology, its poor overall prognosis, and its modest response to therapeutic interventions. Between 30,000 and 40,000 patients are diagnosed with IPF yearly in the US [60]. IPF has a median survival of approximately three years in most published studies, which is worse than many malignancies in the 21^st^ century [60]. 

Despite decades of intense research and exploration of numerous potential causes, the etiology and precise pathobiology of IPF remains unknown. We mentioned earlier in this review that in patients with ILD, treatment response and clinical prognosis are likely determined more by the etiology of ILD and the temporal nature of the lung injury (ceased versus persistent) rather than a specific histopathologic pattern. In IPF, the etiology of lung injury is unknown, but it seems very likely that the cause of lung injury is unable to be eliminated, and thus molecular mechanisms of injury response and lung fibrosis as a result of such injury are thus ongoing and persistent. 

Despite its importance, the term IPF, which was coined in the 1970s, may be misleading and potentially confusing. Dissecting the individual words, idiopathic pulmonary fibrosis states that the patient has pulmonary fibrosis of unknown cause. This connotation would imply a very broad umbrella-type term, which could be associated with numerous radiologic and histopathologic patterns. However, over the past 10–15 years, the term IPF in clinical pulmonary medicine, research studies and consensus statements has been restricted to a narrow and well-defined phenotype of disease, with specific clinical, radiologic, and histopathologic findings [4,5,6]. 

Accurately identifying a patient with IPF is based on a comprehensive medical history, physical examination, strict radiologic criteria, and strict histopathologic criteria, if lung biopsy is obtained [1,2,4,6]. This IPF phenotype is distinct from the other ILD groups previously discussed. IPF is generally a disease of middle-to older-aged men, with typical age of presentation in the range of 60–70 years, and with men carrying 80–90% of the disease burden in most epidemiologic observations [60]. The majority of patients with IPF report a history of tobacco smoking [169]. In IPF, a comprehensive environmental, avocational, occupational, and medication use history does not identify a plausible etiology of ILD [4,6,7] and there should be no evidence of systemic CTD by history, physical exam, and autoimmune serologic testing. 

The radiographic findings used to support the diagnosis of IPF in the appropriate clinical context are strictly defined (Figure 4) [4,6]. Reticular opacities with or without honeycombing must be present in a peripheral-, posterior-, and basilar-predominant distribution. The peripheral opacities are present in the extreme periphery of the lung, juxtaposed to the visceral pleura, without evidence of subpleural sparing which may be seen in other ILD disorders. Honeycombing in subpleural and basilar distributions is characteristic of IPF and is thus one of its defining features. GGO, mosaic attenuation, nodules, and consolidation should be minimal to absent on chest CT imaging in IPF [4,6,170]. It is noteworthy that patients with IPF are at increased risk for development of small cell and non-small cell lung cancer [171]. Therefore, the appearance of new pulmonary parenchymal findings or growth in size of prior findings on follow-up CT imaging in patients with IPF should be carefully evaluated in that context.

In a patient with the typical clinical and radiographic findings for IPF, a lung biopsy is unlikely to provide additional helpful information, is associated with some risk of patient deterioration [34], and is thus most often not indicated [2,4,6]. If lung biopsy is performed in selected instances or if a lung explant is available following lung transplantation, the histopathologic pattern that supports the diagnosis of IPF is UIP [39]. 

Therapy for IPF for many years has been very poor. Corticosteroids and other immunosuppressive therapies have consistently been demonstrated to have little if any therapeutic efficacy [172,173,174,175] and at times have been demonstrated to be harmful [176,177]. Over the past few years, two new anti-fibrotic therapies—pirfenidone and nintedanib—have been approved by the FDA for treatment of patients with IPF. Pirfenidone is an oral anti-fibrotic agent with some anti-inflammatory properties [178], and nintedanib is a tyrosine kinase inhibitor targeting platelet-derived, fibroblast, and vascular endothelial growth factor receptors [179]. Both pirfenidone and nintedanib have been shown to modestly reduce the decline in lung function over time in patients with IPF, although improvements in overall survival have not been demonstrated to date [180,181,182,183]. Side effects with these two medications may be substantial and thus careful decision making with individual patients weighing benefits versus risks is appropriate [5,181,184].

IPF often develops without much warning in otherwise rather healthy adults, has a very poor prognosis, and has a modest response to currently available therapies. Consequently, in addition to consideration of therapy with pirfenidone or nintedanib, early referral for lung transplantation evaluation should be strongly considered in patients with IPF [4,5,6]. Thus, accurately identifying the IPF phenotype clinically and radiographically and distinguishing patients with IPF from one of the other forms of potentially more treatable and remediable forms of ILD has paramount importance.

## 8. Conclusions

In this review, we have discussed general principles of ILD and pulmonary fibrosis, have emphasized the importance of a comprehensive avocational, environmental, occupational, and medication-use history, have hopefully provided clarity on current clinical, radiologic and histopathologic nomenclature, and have discussed six of the most common and important forms of ILD: Smoking-related ILD, HP, CTD-ILD, occupation-related ILD, medication-induced ILD, and IPF. Many other less common forms of ILDs do exist, but were not discussed due to the scope of this review. Some examples include cystic forms of ILD, pulmonary eosinophilic syndromes, pulmonary alveolar proteinosis, graft-versus-host disease, and ILD associated with systemic inheritable disorders. Discussions of many of these disorders have been comprehensively reviewed elsewhere [185,186,187,188]. Post-infectious ILD is often overlooked as a common cause of radiographic or histopathologic findings which suggest an OP pattern of disease, but we believed this was beyond our scope. Pleuro-parenchymal fibroelastosis (PPFE) is a rare form of ILD that has gained more recognition over the past ten years. Upper lobe pleural-based findings are characteristic and PPFE has been described as idiopathic disease or associated with familial or autoimmune processes [2]. Sarcoidosis is another very important ILD, but in our experience, its phenotype is often better understood by general medicine and pulmonary physicians alike, and thus we chose to not include discussion of sarcoidosis in this review. 

We aimed to guide the general medicine physician in arriving at a precise ILD diagnosis, to consider efforts regarding environmental remediation or cessation of harmful exposures, to consider institution of anti-inflammatory, immunosuppressive, or anti-fibrotic therapies where appropriate, and to identify potentially pulmonary-toxic therapies when pertinent. Additionally, physicians should consider a pulmonary rehabilitation program which can often help ILD patients understand their disease process and improve their quality of life, and all patients with ILD should be assessed at rest and during ambulation for evidence of oxygen desaturation and possible requirement for supplemental home oxygen. In selected instances, patients with progressive disease may be considered good candidates for lung transplantation, and referral to a physician with transplantation expertise would be appropriate. Unfortunately, in some instances such as advanced IPF, ILD may be a progressive and poorly treatable disease despite the best efforts of physicians, and in these situations, palliative counseling and utilization of palliative care resources would be appropriate.

Arriving at a precise and accurate diagnosis of the etiology of ILD requires integration of all pertinent clinical information available: Medical history and physical exam findings, autoimmune serologic testing results, chest CT imaging findings, and histopathologic findings on lung biopsy, if available. Discussion amongst the general medicine physician, pulmonologist, chest radiologist, pulmonary pathologist, and in some cases rheumatologist, in the context of a multidisciplinary setting is often helpful. Clinical prognosis and response to immunosuppressive therapies are more likely to be determined by the etiology of ILD than any particular radiologic or histopathologic pattern. If there is one overall message that we would like to convey, it is that a careful and detailed environmental, avocational, occupational, and medication-use history is likely the major factor which will lead to an accurate diagnosis in ILD and to hopefully achieve the best possible outcomes for patients.

## Figures and Tables

**Figure 1 jcm-07-00476-f001:**
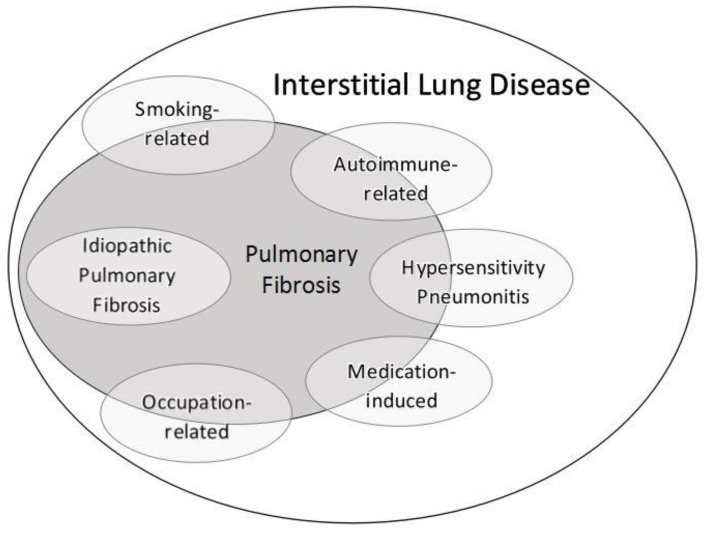
Schematic overview of interstitial lung disease (ILD) and pulmonary fibrosis, which includes the six ILDs discussed in this review. Other less common ILDs would occupy remaining space within the ILD and fibrosis ovals and would likewise manifest varying degrees of inflammation and/or fibrosis depending on the particular disorder.

**Figure 2 jcm-07-00476-f002:**
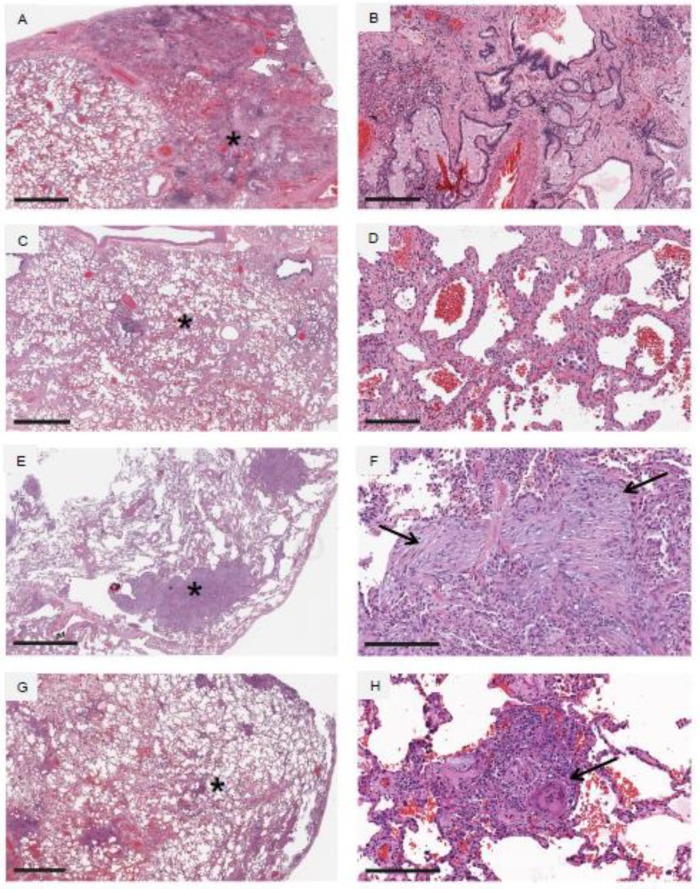
Common histopathologic patterns observed in various forms of ILD. Each pair of horizontal panels represent low (left panel) and high (right panel) magnification images taken from the same tissue section, and the high magnification images are taken from areas identified by an asterisk (*) on low magnification. Panels (**A**,**B**) demonstrate usual interstitial pneumonia (UIP). Low magnification demonstrates dense fibrotic change and destruction of normal lung architecture on the right side of panel (**A**), and relatively normal lung on the left side of panel (**A**). There are widespread areas of honeycomb change observed in the fibrotic areas, as seen on low and high magnification. Panels (**C**,**D**) demonstrate non-specific interstitial pneumonia (NSIP). Low magnification demonstrates uniform thickening of the alveolar walls (i.e., the lung interstitium) diffusely throughout the lung, and high magnification demonstrates a combination of cellular inflammation (lymphocytes and macrophages) as well as collagen deposition in these thickened walls. Panels (**E**,**F**) demonstrate organizing pneumonia (OP). Low magnification demonstrates preservation of overall lung architecture, but large focal areas of lung abnormality (asterisk in panel (**E**)) are observed; high magnification demonstrates these focal abnormalities to be areas of organization (arrows in panel (**F**)). Panels (**G**,**H**) demonstrate granulomatous inflammation. Low magnification demonstrates widespread small focal areas of inflammation centered around small airways, and high magnification demonstrates many of these focal areas of inflammation to contain poorly formed granulomas and multinucleated giant cells (arrow in panel **H**). Horizontal black bars in the lower left of each of the panels (**A**–**H**) represent 2 mm (panels **A**,**C**,**E**,**G**), 300 µm (panel **B**), and 200 µm (panels **D**,**F**,**H**).

**Figure 3 jcm-07-00476-f003:**
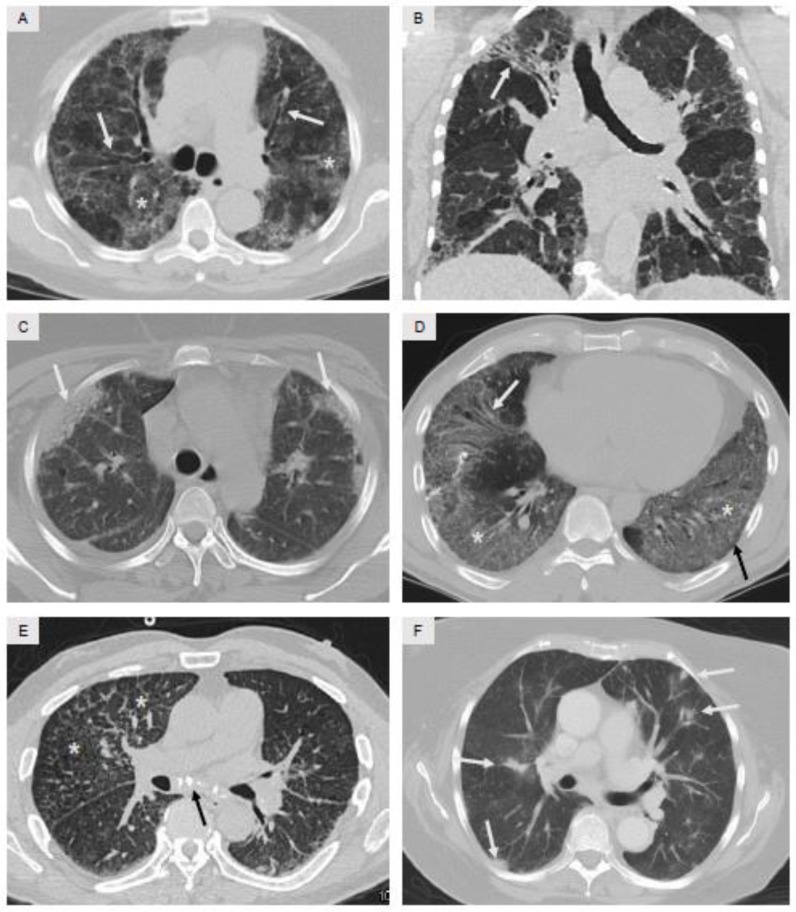
Common radiologic patterns observed on chest CT imaging in various forms of ILD. Panel (**A**) demonstrates widespread GGO (asterisk) and accompanying traction bronchiectasis (arrows); overall clinical diagnosis was smoking-related ILD. Panel (**B**) demonstrates findings of fibrosis in the right upper lobe with traction bronchiectasis and volume loss (arrow), and widespread areas of bilateral mosaic attenuation, defined as interspersed areas of hyperattenuated lung (gray) and hypoattenuated lung (black); overall clinical diagnosis was hypersensitivity pneumonitis. Panel (**C**) demonstrates peripheral areas of consolidation (arrows); overall clinical diagnosis was ILD secondary to systemic lupus erythematosus (SLE). Panel (**D**) demonstrates widespread areas of GGO (asterisk), mild sparing of the extreme periphery of the lung (black arrow), and mild traction bronchiectasis (white arrow); overall clinical diagnosis was ILD secondary to anti-synthetase syndrome. Panel **E** demonstrates small nodules, reticular opacities, and prominent interlobular septa in the right upper lobe (asterisk) and to a lesser extent in the left lower lobe, along with calcified mediastinal adenopathy (arrow); overall clinical diagnosis was ILD secondary to chronic beryllium exposure. Panel **F** demonstrates multiple ill-defined nodules (arrows) along with surrounding GGO; overall clinical diagnosis was ILD secondary to amiodarone.

**Figure 4 jcm-07-00476-f004:**
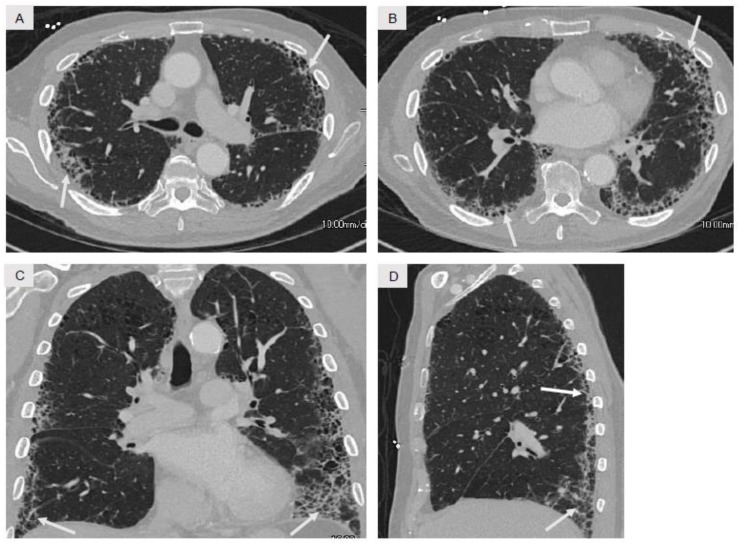
Typical radiologic findings observed on chest CT imaging in idiopathic pulmonary fibrosis (IPF), with axial (**A**,**B**) coronal (**C**), and sagittal (**D**) images. The nature of the opacities (curvilinear reticular opacities and honeycomb change) and their distribution (markedly peripheral and lower lobe-predominant) are very consistent with IPF. Note the absence of subpleural sparing, GGO, consolidation, mosaic attenuation, and nodules which are often observed in other forms of ILD as shown in Figure 3.

**Table 1 jcm-07-00476-t001:** Distinguishing features among six common ILD subgroups.

ILD Subgroup	Medical History Clues	Radiographic Findings	Histologic Patterns	Autoimmune Serology Evaluation	Therapeutic Considerations
Smoking-Related	Prior or current tobacco smoking, biomass combustion exposure	GGO, emphysema with fibrosis, mosaic attenuation	RB, DIP, PLCH, emphysema, fibrosis	Normal or non-reactive	Smoking cessation, removal from exposure, anti-inflammatory therapy in advanced disease
Hypersensitivity Pneumonitis	Exposure to environmental organic antigens (avian proteins, mold, farming-related)	GGO, mosaic attenuation, upper-lobe fibrosis, centrilobular nodules	Poorly-formed granulomas with multinucleated giant cells	Normal or non-reactive; consider serum precipitating antibody panel	Removal from exposure, remediation of environment, anti-inflammatory and/or immunosuppressive therapy in advanced disease
Connective Tissue Disease-Related	Constitutional symptoms, arthralgia/arthritis, rash, Raynaud’s phenomenon	GGO, reticular opacities, consolidation, fibrosis, peripheral sparing	Chronic inflammation, UIP, NSIP, or OP	Autoimmune evaluation recommended	Anti-inflammatory and/or immunosuppressive therapy
Occupation-Related	Current or prior at-risk occupation	GGO, reticular opacities, fibrosis, nodules, pleural plaques (asbestos)	Chronic inflammation, UIP, NSIP, OP, or granulomatous inflammation	Normal or non-reactive; lymphocyte proliferation test if beryllium exposure suspected	Removal from exposure, use of respiratory protective equipment
Medication-Induced	Current or prior use of possible offending medications or therapies	GGO, reticular opacities, consolidation, nodules, fibrosis	Chronic inflammation, UIP, NSIP, OP, or granulomatous inflammation	Usually normal or non-reactive	Medication discontinuation if possible, anti-inflammatory and/or immunosuppressive therapy in advanced disease
Idiopathic Pulmonary Fibrosis	Male predominance, 7th–8th decade, past history of tobacco smoking or dust exposure	Peripheral and basilar fibrosis with honeycombing (Figure 4)	UIP	Normal or non-reactive	Consideration of antifibrotic therapies, lung transplantation evaluation in selected patients

GGO, ground-glass opacities; RB, respiratory bronchiolitis; DIP, desquamative interstitial pneumonia; PLCH, pulmonary Langerhans cell histiocytosis; UIP, usual interstitial pneumonia; NSIP, nonspecific interstitial pneumonia; OP, organizing pneumonia.

**Table 2 jcm-07-00476-t002:** Suggested serum serologies in the evaluation of patients with ILD.

**Serum Markers of Systemic Inflammation**			
● erythrocyte sedimentation rate (ESR)			
● c-reactive protein (CRP)			
**Autoantibodies**			
Anti-neutrophil cytoplasmic antibodies (ANCAs)			
● c-ANCA, p-ANCA, atypical p-ANCA			
● anti-MPO			
● anti-PR3			
Rheumatoid arthritis			
● anti-CCP			
● rheumatoid factor (RF)			
SLE and related autoimmune diseases (MCTD, Sjögren’s syndrome)			
● ANA			
● anti-dsDNA			
● anti-SS-A (Ro)			
● anti-SS-B (La)			
● anti-Smith			
● anti-RNP			
Systemic sclerosis			
● anti-Scl-70			
● anti-Centromere			
Myositis-specific and myositis-associated antibodies			
● anti-Jo1	● anti-PL7	● anti-PL12	● anti-EJ
● anti-OJ	● anti-SRP	● anti-Mi-2α	● anti-Mi-2β
● anti-MDA5	● anti-TIF-1γ	● anti-NXP-2	● anti-KS
● anti-Zo	● anti-Ku		
**Muscle Enzymes**			
● CPK			
● aldolase			

Abbreviations: c, cytoplasmic; p, perinuclear; MPO, myeloperoxidase; PR3, proteinase 3; CCP, cyclic citrullinated peptide; SLE, systemic lupus erythematosus; ANA, anti-nuclear antibody; dsDNA, double stranded DNA; SS-A, Sjögren’s syndrome-related antigen A; SS-B, Sjögren’s syndrome-related antigen B; RNP, ribonucleoprotein; MDA, melanoma differentiation-associated gene; SRP, signal recognition particle; TIF, transcriptional intermediary factor; NXP, nuclear matrix protein; CPK, creatine phosphokinase.

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
