# Peer review of "Interstitial Lung Disease and Pulmonary Fibrosis: A Practical Approach for General Medicine Physicians with Focus on the Medical History"

_jcm, 2018, doi:10.3390/jcm7120476_

Reviewer 1 Report

This review is aimed at the non-specialist, focusing on the importance of the medical history in determining the cause of ILD.

It is eloquent and provides a concise overview of the topic.

I only have a few comments:

Title:

I would change this to make it apply to the non-American audience - perhaps changing internist to non-specialist or general medical physician?

Section on histology :

-Fibrotic foci are an important histological component of UIP and should be mentioned.

-granulomatous inflammation: perhaps mention also feature of sarcoid (not covered in the review)

-lines 173/174: comment that UIP in CTD-ILD often have impressive improvements in pulmonary status and lung function - I would be more reserved in the wording here - UIP pattern in CD-ILD still confers a worse prognosis than other histological types.

Hypersensitivity pneumonitis section: 

- Antigen is commonly not-identified (quoted as high as 60% in some papers) - these needs to be highlighted.

- Perhaps also highlight that some retrospective analysis such that the response to therapy might be predicted depending on presenting radiological features (Chest 2018).

Connective tissue disease section:

- Table not included.

Occupational lung disease:

-Highlight that radiologically asbestosis can be indistinguishable from IPF.

General comment:

- In each section a comment has been made about the treatments to be considered. I would like to see a general comment with regards to pulmonary rehabilitation/oxygen assessments and palliation (particularly for IPF).

Author Response

November 21, 2018

 Xu Yang

Assistant Editor, Journal of Clinical Medicine

 Alicia Ren

Assistant Editor, Journal of Clinical Medicine

 Re:       Interstitial Lung Disease and Pulmonary Fibrosis: A Practical Approach for Internists with Focus on the Medical History

Journal: Journal of Clinical Medicine

Manuscript ID: jcm-396229

 Dear Journal of Clinical Medicine Editorial Team:

 Thank you very much for inviting resubmission of our manuscript initially entitled “Interstitial Lung Disease and Pulmonary Fibrosis: A Practical Approach for General Medicine Physicians with Focus on the Medical History.” We also thank the Editors and Reviewers for their careful evaluation of the manuscript, and for their insightful recommendations. We have addressed each of the specific suggestions and comments in our outlined point-by-point responses below, and have made the recommended changes in the attached manuscript using track-changes mode as suggested. We hope that our manuscript is now acceptable for publication in the Journal of Clinical Medicine.

 Reviewer 1

We thank the Reviewer for the careful and detailed attention to our manuscript and for the overall very remarks. We are grateful for the insightful thoughts and recommended suggestions. The suggested recommendations have been implemented, and are found in the revised manuscript using the track-changes mode as suggested, and as documented below. 

 1.       Title: I would change this to make it apply to the non-American audience - perhaps changing internist to non-specialist or general medical physician?

As recommended, we have changed the title of the manuscript from internist to General Medical Physician, and amended the terminology accordingly in the abstract and manuscript.

 2.       Section on histology: Fibrotic foci are an important histological component of UIP and should be mentioned.

As recommended, we have amended the histology section on UIP to include fibroblastic foci.

Granulomatous inflammation: perhaps mention also feature of sarcoid (not covered in the review).

As recommended, we have amended the histology section to include granulomatous inflammation as a component of sarcoidosis in addition to hypersensitivity pneumonitis.

Lines 173/174: comment that UIP in CTD-ILD often have impressive improvements in pulmonary status and lung function - I would be more reserved in the wording here - UIP pattern in CD-ILD still confers a worse prognosis than other histological types. 

As recommended, we have amended our language for this sentence to be more reserved and be cautious with our wording, and have included an appropriate reference.

3.       Hypersensitivity pneumonitis section: Antigen is commonly not-identified (quoted as high as 60% in some papers) - these needs to be highlighted. 

As recommended, we have amended the language in the HP section to clarify that antigen identification with suspected HP may be difficult, and have included an appropriate reference

Perhaps also highlight that some retrospective analysis such that the response to therapy might be predicted depending on presenting radiological features (Chest 2018). 

As recommended, we have amended the manuscript to indicate that response to therapy in HP may be predicted based on presenting radiologic features.

4.       Connective tissue disease section: Table not included. 

We apologize that the Reviewer was unable to view our Table related to our suggested serologic evaluation of patients with CTD–ILD.  We have uploaded this as Table 2 with the revised manuscript, and hope it is now visible to the Reviewer.

 5.       Occupational lung disease: Highlight that radiologically asbestosis can be indistinguishable from IPF. 

As recommended, we have amended the manuscript to highlight the similarities between asbestosis and IPF.

6.       General comment: In each section a comment has been made about the treatments to be considered. I would like to see a general comment with regards to pulmonary rehabilitation/oxygen assessments and palliation (particularly for IPF).

As recommended, a new paragraph regarding pulmonary rehabilitation, assessment for oxygen need, and appropriate use of palliative care resources has been added to the Conclusion.

Reviewer 2

We thank the Reviewer for the very encouraging comments on our manuscript, and for the insightful thoughts and recommendations. The recommendations suggested have been implemented, and are found in the revised manuscript using the track-changes mode as suggested, and as documented below. 

 1.       Family history taking and possible genetic predispositions (familial pulmonary fibrosis and gene polymorphisms).

As recommended by the Reviewer, a new paragraph on the importance of family history and potential genetic contributions has been added to the Comprehensive History section

 2.       Pleuroparenchymal fibroelastosis (PPFE).

As recommended, we have added pleuroparenchymal fibroelastosis (PPFE) to the Conclusion, with an appropriate reference.

 3.       Radiation pneumonitis and recall pneumonitis.

As recommended, we have added a discussion radiation pulmonary toxicity to the Medication-induced section, with appropriate references

 4.       Pulmonary infection that resembles ILD  (Particularly CMV and pneumocystis under the immunosuppressive treatment).

As recommended, we added a new paragraph within the Comprehensive History to consider the possibility of a diffuse pulmonary infectious process.

5.       Lung cancer as a major comorbidity of IPF.

As recommended, we added a discussion on lung cancer in the IPF section.

 6.       A table summarizing the six subtypes of ILD would be helpful for the readers (diagnostic clues, radiologic/histologic/serologic characteristics, and treatments). 

As recommended, a new Table 1 has been created, and has been uploaded with the revised manuscript.

 7.       Page 10, Line 268: No table is shown. Page 11, Line 399: No table is shown.

We apologize that our Table related to suggested serologic evaluation of patients with CTD–ILD was not able to be viewed by the Reviewers. We have uploaded this as Table 2 with the revised manuscript, and we hope that this Table is now visible to the Reviewer.

 8.       I noticed several typing errors and I recommend careful proof-reading. For example, Line 21: lack of spacing, Line: 129 references, Line 255: “wide spread”, Line 285: double spacing?, Line 480: "programmed cell death-1”.

As recommended, we have carefully proof-read the manuscript again, and have corrected the I identified deficiencies.

Reviewer 2 Report

This review presents a well-organized guide for the management of ILD, which assists clinical practice of general physicians and pulmonologists. This manuscript also incorporates recent topics on the emerging clinical problems, concepts, and novel therapeutics. I feel that the authors could also mention some of the following points:

Family history taking and possible genetic predispositions (familial pulmonary fibrosis and gene polymorphisms)

Pleuroparenchymal fibroelastosis (PPFE)

Radiation pneumonitis and recall pneumonitis

Pulmonary infection that resembles ILD

(Particularly CMV and pneumocystis under the immunosuppressive treatment)

Lung cancer as a major comorbidity of IPF

 A table summarizing the six subtypes of ILD would be helpful for the readers (diagnostic clues, radiologic/histologic/serologic characteristics, and treatments).

 Page 10, Line 268: No table is shown.

Page 11, Line 399: No table is shown.

 I noticed several typing errors and I recommend careful proof-reading.

For example,

Line 21: lack of spacing

Line: 129 references

Line 255: “wide spread”

Line 285: double spacing?

Line 480: "programmed cell death-1”

Author Response

November 21, 2018

Xu Yang

Assistant Editor, Journal of Clinical Medicine

Alicia Ren

Assistant Editor, Journal of Clinical Medicine

Re: Interstitial Lung Disease and Pulmonary Fibrosis: A Practical Approach for Internists with Focus on the Medical History

Journal: Journal of Clinical Medicine

Manuscript ID: jcm-396229

Dear Journal of Clinical Medicine Editorial Team:

Thank you very much for inviting resubmission of our manuscript initially entitled “Interstitial Lung Disease and Pulmonary Fibrosis: A Practical Approach for General Medicine Physicians with Focus on the Medical History.” We also thank the Editors and Reviewers for their careful evaluation of the manuscript, and for their insightful recommendations. We have addressed each of the specific suggestions and comments in our outlined point-by-point responses below, and have made the recommended changes in the attached manuscript using track-changes mode as suggested. We hope that our manuscript is now acceptable for publication in the Journal of Clinical Medicine.

Reviewer 1

We thank the Reviewer for the careful and detailed attention to our manuscript and for the overall very remarks. We are grateful for the insightful thoughts and recommended suggestions. The suggested recommendations have been implemented, and are found in the revised manuscript using the track-changes mode as suggested, and as documented below. 

1. Title: I would change this to make it apply to the non-American audience - perhaps changing internist to non-specialist or general medical physician?

As recommended, we have changed the title of the manuscript from internist to General Medical Physician, and amended the terminology accordingly in the abstract and manuscript.

2. Section on histology: Fibrotic foci are an important histological component of UIP and should be mentioned.

As recommended, we have amended the histology section on UIP to include fibroblastic foci.

Granulomatous inflammation: perhaps mention also feature of sarcoid (not covered in the review).

As recommended, we have amended the histology section to include granulomatous inflammation as a component of sarcoidosis in addition to hypersensitivity pneumonitis.

Lines 173/174: comment that UIP in CTD-ILD often have impressive improvements in pulmonary status and lung function - I would be more reserved in the wording here - UIP pattern in CD-ILD still confers a worse prognosis than other histological types.

As recommended, we have amended our language for this sentence to be more reserved and be cautious with our wording, and have included an appropriate reference.

3. Hypersensitivity pneumonitis section: Antigen is commonly not-identified (quoted as high as 60% in some papers) - these needs to be highlighted.

As recommended, we have amended the language in the HP section to clarify that antigen identification with suspected HP may be difficult, and have included an appropriate reference

Perhaps also highlight that some retrospective analysis such that the response to therapy might be predicted depending on presenting radiological features (Chest 2018).

As recommended, we have amended the manuscript to indicate that response to therapy in HP may be predicted based on presenting radiologic features.

4. Connective tissue disease section: Table not included.

We apologize that the Reviewer was unable to view our Table related to our suggested serologic evaluation of patients with CTD–ILD. We have uploaded this as Table 2 with the revised manuscript, and hope it is now visible to the Reviewer.

5. Occupational lung disease: Highlight that radiologically asbestosis can be indistinguishable from IPF.

As recommended, we have amended the manuscript to highlight the similarities between asbestosis and IPF.

6. General comment: In each section a comment has been made about the treatments to be considered. I would like to see a general comment with regards to pulmonary rehabilitation/oxygen assessments and palliation (particularly for IPF).

As recommended, a new paragraph regarding pulmonary rehabilitation, assessment for oxygen need, and appropriate use of palliative care resources has been added to the Conclusion.

Reviewer 2

We thank the Reviewer for the very encouraging comments on our manuscript, and for the insightful thoughts and recommendations. The recommendations suggested have been implemented, and are found in the revised manuscript using the track-changes mode as suggested, and as documented below.

1. Family history taking and possible genetic predispositions (familial pulmonary fibrosis and gene polymorphisms).

As recommended by the Reviewer, a new paragraph on the importance of family history and potential genetic contributions has been added to the Comprehensive History section

2. Pleuroparenchymal fibroelastosis (PPFE).

As recommended, we have added pleuroparenchymal fibroelastosis (PPFE) to the Conclusion, with an appropriate reference.

3. Radiation pneumonitis and recall pneumonitis.

As recommended, we have added a discussion radiation pulmonary toxicity to the Medication-induced section, with appropriate references

4. Pulmonary infection that resembles ILD (Particularly CMV and pneumocystis under the immunosuppressive treatment).

As recommended, we added a new paragraph within the Comprehensive History to consider the possibility of a diffuse pulmonary infectious process.

5. Lung cancer as a major comorbidity of IPF.

As recommended, we added a discussion on lung cancer in the IPF section.

6. A table summarizing the six subtypes of ILD would be helpful for the readers (diagnostic clues, radiologic/histologic/serologic characteristics, and treatments).

As recommended, a new Table 1 has been created, and has been uploaded with the revised manuscript.

7. Page 10, Line 268: No table is shown. Page 11, Line 399: No table is shown.

We apologize that our Table related to suggested serologic evaluation of patients with CTD–ILD was not able to be viewed by the Reviewers. We have uploaded this as Table 2 with the revised manuscript, and we hope that this Table is now visible to the Reviewer.

8. I noticed several typing errors and I recommend careful proof-reading. For example, Line 21: lack of spacing, Line: 129 references, Line 255: “wide spread”, Line 285: double spacing?, Line 480: "programmed cell death-1”.

As recommended, we have carefully proof-read the manuscript again, and have corrected the identified deficiencies.